# The Arising Role of Extracellular Vesicles in Cholangiocarcinoma: A Rundown of the Current Knowledge Regarding Diagnostic and Therapeutic Approaches

**DOI:** 10.3390/ijms242115563

**Published:** 2023-10-25

**Authors:** Eleni-Myrto Trifylli, Anastasios G. Kriebardis, Evangelos Koustas, Nikolaos Papadopoulos, Sofia Vasileiadi, Sotirios P. Fortis, Vassilis L. Tzounakas, Alkmini T. Anastasiadi, Panagiotis Sarantis, Effie G. Papageorgiou, Ariadne Tsagarakis, Georgios Aloizos, Spilios Manolakopoulos, Melanie Deutsch

**Affiliations:** 1Laboratory of Reliability and Quality Control in Laboratory Hematology (HemQcR), Department of Biomedical Sciences, Section of Medical Laboratories, School of Health & Caring Sciences, University of West Attica (UniWA), Ag. Spyridonos Str., 12243 Egaleo, Greece; trif.lena@gmail.com (E.-M.T.); sfortis@uniwa.gr (S.P.F.); efipapag@uniwa.gr (E.G.P.); 2First Department of Internal Medicine, 417 Army Share Fund Hospital, 11521 Athens, Greece; aloizosgio@yahoo.gr; 32nd Academic Department of Internal Medicine, Medical School, National and Kapodistrian University of Athens, Hippokration General Hospital of Athens, Vasilissis Sofias Avenue Str., 11527 Athens, Greece; vasileiadisofia@gmail.com (S.V.); smanolak@med.uoa.gr (S.M.); meladeut@gmail.com (M.D.); 4Oncology Department, General Hospital Evangelismos, 10676 Athens, Greece; vang.koustas@gmail.com; 5Department of Biological Chemistry, Medical School, National and Kapodistrian University of Athens, 11527 Athens, Greece; panayotissarantis@gmail.com; 6Second Department of Internal Medicine, 401 General Military Hospital, 115 27 Athens, Greece; nipapmed@gmail.com; 7Department of Biochemistry, School of Medicine, University of Patras, 26504 Patras, Greece; vtzounakas@upatras.gr (V.L.T.); aanastasiadi@upatras.gr (A.T.A.); 8Beth Israel Deaconess Medical Center, Harvard Medical School, Boston, MA 02215, USA; a.tsagarakis96@gmail.com

**Keywords:** extracellular vesicles, cholangiocarcinoma, biomarkers, drug delivery, exosomes, microvesicles, therapy

## Abstract

Cholangiocarcinomas (CCAs) constitute a heterogeneous group of highly malignant epithelial tumors arising from the biliary tree. This cluster of malignant tumors includes three distinct entities, the intrahepatic, perihilar, and distal CCAs, which are characterized by different epidemiological and molecular backgrounds, as well as prognosis and therapeutic approaches. The higher incidence of CCA over the last decades, the late diagnostic time that contributes to a high mortality and poor prognosis, as well as its chemoresistance, intensified the efforts of the scientific community for the development of novel diagnostic tools and therapeutic approaches. Extracellular vesicles (EVs) comprise highly heterogenic, multi-sized, membrane-enclosed nanostructures that are secreted by a large variety of cells via different routes of biogenesis. Their role in intercellular communication via their cargo that potentially contributes to disease development and progression, as well as their prospect as diagnostic biomarkers and therapeutic tools, has become the focus of interest of several current studies for several diseases, including CCA. The aim of this review is to give a rundown of the current knowledge regarding the emerging role of EVs in cholangiocarcinogenesis and their future perspectives as diagnostic and therapeutic tools.

## 1. Introduction

Cholangiocarcinoma (CCA) constitutes a highly aggressive and heterogeneous cluster of epithelial tumors located along the biliary tree, giving rise to three different entities of malignant tumors based on their anatomical site [1]. In the intrahepatic part of the biliary tree, two types of tumors can be identified, including the (i) intrahepatic CCA (iCCA) and another rare form of a mixed tumor, the so-called hepatocellular (CHC-CCA) that is originated from transdifferentiated hepatocytes [2,3]. The iCCA is considered the second most commonly diagnosed primary hepatic malignancy after hepatocellular carcinoma (HCC), presenting several genetic alterations, such as *FGFR* and *IDH1/2* fusion, as well as *ARID1A* and *BRAF* mutations [3]. The second distinct entity is (ii) perihilar CCA (pCCA), which is the most frequently diagnosed CCA form, presenting approximately in 50–60% of CCA patients [4]. The third form is (iii) distal CCA (dCCA), the second most common form of CCA (20–30% of CCA cases), while the two extrahepatic CCA types commonly present a different molecular background than iCCA, including *PRKACA-PRKACB* fusions and mutations of *ELF3* and *ERBBE* genes [4,5]. Meanwhile, the aggressive course of this malignancy is closely associated with *Tp53* and *KRAS* mutations, with the former being closely related to viral hepatitis B infection, which commonly co-occurs [6]. Last but not least, some other mutations are identified and possibly availed for therapeutic targeting, such as *SMAD4* and *PPHLN1*, as well as *ROS*, *MGEA5*, *BICC1*, *TACCE*, and epigenetic aberrations such as those in microRNAs expression levels and DNA hyper-methylation [7].

The heterogeneity is also identified on a geographical level and ethnical level, implying the significant role of local risk factors in combination with genetic and sex factors, exhibiting a male predominance [8]. There is a noticeable increase of CCA incidence in Western countries over the last years, a phenomenon that is mainly attributed to the chronic inflammation of biliary ducts [9]. Chronic inflammation of the biliary tree is presented in different diseases such as primary sclerosing cholangitis (PSC), which constitutes the most well-studied risk factor, viral hepatitis B and C (HBV and HCV), type 2 diabetes mellitus, metabolic (dysfunction)-associated fatty liver disease (MAFLD), and lithiasis, as well as chronic pancreatitis, cirrhosis, and the heritable fibro polycystic disease [9,10]. Meanwhile, it is observed that the highest incidence of CCA is situated in Eastern countries, which is mainly attributed to food contamination with liver fluke larvae of the *Clonorchis sinensis* and *Opisthorchis viverrini* species, as well as to professional exposure to different substances, such as aflatoxins [11]. 

Despite the fact that CCA is considered to not be a frequent gastrointestinal malignancy (3% of cases), the increasing incidence, as well as the late diagnostic time that contributes to high mortality rates, is among the reasons for the intensified efforts for the development of novel biomarkers as diagnostic tools and therapeutic approaches [12]. 

However, the late diagnostic time is not the only limitation. There are several other obstacles in the management of CCA patients, including the selection of patients who are suitable for surgical treatment, depending on their health status, the tumor site, and the stage of the disease; while orthotopic liver transplantation is still controversial in many countries. There also several restrictions for loco regional treatment, including photodynamic therapy (PDT), conventional or drug-eluting transarterial chemoembolization (TACE), and radiofrequency ablation, such as the anatomical complexity of the biliary tree and the possible radiation-associated side effects in the adjacent organs. Despite the advances in the medical treatment options, including chemotherapy, immunotherapy, and targeted therapies, CCAs still remain resistant to treatment, due to the fact that they are quite heterogeneous, requiring a personalized approach for each patient based on the epigenetic, genetic, and epidemiological background of the tumor. Another major challenge is the differentiation of iCCA from HCC based on the current imaging methods, which is an issue that commonly requires the performance of an invasive tumor biopsy. Similarly, there is another diagnostic dilemma for the differentiation of malignant from benign bile duct strictures, which requires the approach of the biliary tree by a skilled operator, who should perform demanding examinations such as percutaneous transhepatic cholangiography or endoscopic retrograde cholangiopancreatography (ERCP) or single-operator peroral cholangioscopy (SOP), with the collected cytological samples or serum CA19-9 biomarker having low sensitivity [1,2,4,12].

All the aforementioned limitations in the current diagnostic and therapeutic modalities, shifted the interest of the scientific community towards the development of novel non-invasive diagnostic and therapeutic tools, such as extracellular vesicles (EVs), which are in the spotlight of several ongoing studies. These nanosized vesicles are composed of a lipid membrane and they carry several types of cargo that significantly contribute to intercellular communication, while their quantity and quality aberrations are closely implicated in disease development and progression [13]. There is a wide variety of cargoes including different coding and non-coding RNA molecules (short microRNAs (miRs) and long non-coding RNA (lncRNA)), autophagosomes, DNA sequences, proteins, and mitochondrial DNA, as well as lipids, that contribute in different ways in the disease initiation and progress [14]. The interplay between EVs and cholangiocarcinogenesis has gained a considerable amount of attention, especially for the implication of EVs in CCA growth and progression, as well as for their potential use as diagnostic and therapeutic tools [15]. In this review, we will shed light on all the current knowledge about the emerging role of EVs in CCA and its future diagnostic and therapeutic perspectives. 

## 2. An Insight into EVs’ Biogenetic Pathways

EVs constitute heterogenic nanoparticles with a wide variety of sizes that are produced from three distinct biogenetic mechanisms, including (i) the internal budding and (ii) the outward blebbing of the plasma membrane, as well as (iii) cell apoptosis. From the above pathways, exosomes, microvesicles, and apoptotic bodies are released, respectively [16]. The heterogeneity of these nanoparticles is also identified in their cargoes, which are enclosed in their double-lipid membrane. These cargoes can be lipid, DNA, protein molecules, receptors, and autophagosomes, as well as coding and non-coding RNA molecules, including messenger and short or long-non-coding RNA molecules, respectively. The role of the aforementioned cargo is considered quite significant as they are bioactive molecules, which are released from the parental cell and interact with other adjacent or distant cells, establishing crosstalk communication [17]. 

Exosomes are the smallest subclass of EVs (40–150 nm), which are produced via a distinct biogenetic pathway that is initiated with the inward budding of the plasma membrane [17,18]. More particularly, several transmembrane proteins are internalized, under the effect of the endoplasmic network, which is followed by the formation of vesicles that are further split from the cell membrane. Afterward, early endosomes are formed, after being integrated with the aforementioned vesicles, and then are further matured into late endosomes. The internalized molecules can be either transferred back to the plasma membrane or further led to intraluminal vesicles (ILVs). The formation of ILVs includes the invagination of several points of late endosomal membranes, while ILVs additionally form multivesicular bodies (MVBs), under the contribution of an endosomal sorting (ESCRT) complex [18,19]. This enzymatic complex is constituted by ESCRT 0, I–III, which has a pivotal role in the remodeling of the membrane, that finally leads to the formation of MVBs that is composed of several ILVs. More particularly, ESCRT-0 is required for the generation of ILVs, while ESCRT-I is further recruited and allows endosomal membrane budding. In addition, ESCRT-II later recruits ESCRT-III, which allows the separation of ILVs and their incorporation into MVB lumen [19,20]. Nevertheless, there is an alternative ESCRT-independent pathway of exosome generation, when the cargo recruitment arises from divergent sites like the cytoplasm, trans-Golgi complex, and membrane [21]. The last step for the exocytosis of exosomes in the extracellular space is mediated via the MVB integration in plasma membrane, under the involvement of VAMP7, syntaxin 1A (Syx1A), and Ykt6, which constitute a protein complex, the so-called soluble NSF attachment protein receptor (SNARE) proteins [22,23,24]. However, the aforementioned event is not the only pathway that MVBs follow, due to the fact that they can be directed towards lysosomal degradation or can interact with the autophagy pathway [25]. 

Focusing on the medium-sized (150–1000 nm) EVs, the so-called microvesicles, there is a completely different pathway for their biogenesis, as it starts with the outward blebbing of the cell membrane, while it is achieved via the interplay between TSG101 and arrestin-domain-containing protein-1 (ARRDC1) proteins [26]. This outward budding of the cell membrane requires the transfer of TSH101 from endosomal membranes to the former, while the produced microvesicles include TSH101 or ARRDC1 proteins [27]. However, there are several other proteins that take part in membrane blebbing, such as Rab-GTPases and N-ethylmaleimide-sensitive factor attachment proteins (SNAP) receptors (SNARES), which induce cargo recruitment and microvesicle production under hypoxia [28]. For microvesicle biogenetic mechanisms, cargo recruitment is closely regulated by another protein, the so-called ADP-ribosylation factor 6 (ARF6) protein, which induces the selective recruitment of proteins, such as integrins, major histocompatibility complex-I (MHC-I), myosin light-chain kinase (MLCK) proteins, and VAMP3 or nucleic acid molecules [26,29]. 

Furthermore, apoptotic vesicles (ApoEVs) arise from the cell apoptotic pathway and form apoptotic bodies (over 1000 nm) after their fragmentation [30]. More particularly, the biogenetic pathway of the aforementioned entity requires chromatin condensation and the disintegration of the nucleus and intracytoplasmic organelles, resulting in the minimization of cell volume [31]. Additionally, the cell membrane is blebbed with several projections such as microtubule spikes and apoptopodia, as well as beaded apoptopodia, which further lead to apoptotic cell body generation. The latter entity is further fragmented, giving rise to apoptotic bodies [31,32]. 

Meanwhile, there is another pathway of EV generation, which arises from the interplay between autophagy and the exosome biogenetic pathway [33]. More specifically, it has been demonstrated that autophagy is closely associated with EV secretion under stress, while this autophagic pathway is called secretory. This pathway assures cell homeostasis via cargo recruitment and EV release [34], while it is initiated under nutritional indigence, as well as under the disruption of the autophagosome maturation or the autophagosome–lysosome fusion processes [33,34,35]. In addition, autophagy-mediated EV secretion requires autophagosome and MVB fusion, which is closely regulated by the ESCRTs, Rab GTPases, as well as SNAREs, resulting in the formation of the amphisome. This entity is further fused with the lysosome and then integrated with the cell membrane for exosome exocytosis [35]. The orchestration of exosome release is mediated by a number of autophagy-related proteins such as ATG16L1 and ATG5; however, there is another autophagy-related EV secretory pathway that requires only the LC3 conjugation system, the so-called LC3-dependent EV loading and secretion (LDELS) [36].

Last but not least, the role of these bioactive vesicles is pivotal as they significantly participate in cross-talk communication between the parental and the recipient cells [37]. They are delivered to the recipient cells via various endocytic mechanisms such as receptor-associated endocytosis (RME), caveolin-associated endocytosis, micropinocytosis, as well as lipid-mediated endocytosis [38]. Additionally, the delivery method could be via interaction between the ligand and the receptor of the recipient cell. Shedding light on the interactions between the cargoes that are enclosed in the EVs and their role in disease progression, such as CCA, opens up new horizons for the early diagnosis and management of this highly frequent malignant disease (Figure 1) [15,39]. In Figure 1, we demonstrate a schematic presentation of different routes of EV biogenesis (Agreement number UL25ZPCH0P).

## 3. The Role of EVs in Biliary Tract Physiology and Cholangiopathies 

There is a great amount of attention around the role of EVs in biliary tract and liver physiology, as well as on the ways that EVs can be implicated in their pathophysiology [40]. In this section, we will briefly describe the role of EVs in biliary physiological and pathological functions. The first hypothesis that EVs actively participate in biliary functions was made after their identification in bile samples. As was demonstrated, biliary homeostasis is also ensured via the release of EVs, which contain a wide variety of cargoes that interact with several recipient cells [41,42]. 

The polarized cholangiocytes release several EVs from the apical region of the membrane, which is close to the cilia, as well as from the basolateral part. In the former case, the EVs take part in downstream cell–cell communication, towards the biliary tract, while, in the latter, they take part in upstream intercellular communication, towards the hepatocytes and the capillaries around the intrahepatic ducts, which originated from the hepatic artery (Peribiliary capillary plexus, PCP) [43]. An example is the uptake of cholangiocyte-derived EVs by the primary cilia, which constitute biomechanical sensors at the surface of the cells, ensuring the process of several signals from the extracellular space. The role of this phenomenon is the suppression of cell proliferation (ERK-dependent manner), while, once the EV-cargoes are bound to the primary cilia, several signaling pathways are initiated, leading to modifications in the transcriptional level, cell proliferation, differentiation, and bile secretion, as well as in responses to several injurious or inflammatory stimuli [44]. Meanwhile, there are several hepatocyte-derived EVs that are implicated in cholangiocarcinogenesis, such as integrin beta-4 (ITGB4) and epidermal growth factor receptor (EGFR) [43,44]. 

It has to be underlined that EVs that are derived from normal cells in physiological states may have significant quality and quantity aberrations in a disease (inflammatory or malignant), which can potentially induce or enhance inflammatory reactions and fibrotic injuries, in addition to the fact that they are capable of promoting the acquiring of a tumorigenic phenotype by the cells. Shedding light on the fundamental differences between the physiological and the pathological cholangiocyte-derived EVs, we can potentially utilize them as screening tools for patients with PSC, which are prone to develop CCA, based on the proteins that they transfer, which are implicated in carcinogenesis and chronic inflammatory reactions such as in the case of CCA and PSC, respectively [45]. However, further studies are considered necessary for the emergence of the interplay between EVs and biliary tract functions. In Figure 2, we demonstrate a schematic presentation of the apical and basolateral secretion of EVs by polarized cholangiocytes and their targets (Agreement number XU25ZQ4AUV).

## 4. The Implication of EVs in CCA 

As was demonstrated above, several studies demonstrate the significant role of EVs in the regulation of several liver and biliary tract functions, as well as in different types of cholangiopathies, including CCA. In this section, we will describe the interplay between EVs and the CCA-tumor microenvironment (TME), as well as the role of parasite-derived EVs in the induction of the tumorigenic phenotype for cholangiocytes. 

### 4.1. The Interplay between EVs and TME in CCA 

TME constitutes a dynamic system that surrounds the tumor and has a fundamental role in CCA, with a significant implication in CCA progression and metastatic dissemination, as well as neoangiogenesis via different means that will be further discussed [46]. 

#### 4.1.1. A Brief Review of TME Components and Their Role 

This dynamic system is quite heterogeneous, comprising a great variety of cells, an extracellular matrix, and tumor-secreting molecules that lead to chemoresistance and neoangiogenesis, as well as local invasion and distant dissemination. TME consists of several types of cells, including tumor-associated macrophages (TAMs), myeloid-derived suppressor cells (MDSCs), B-regulatory (Breg), natural killer (NK) cells, T-regulatory (Treg) cells, dendritic cells and tumor-infiltrating lymphocytes (TILs), and neutrophils, as well as cancer-associated fibroblasts (CAFs) and endothelial cells (ECs). The innate immune cells of TME, including NK, dendritic, and T cells, as well as TAMs, can either suppress or facilitate tumor development and progression. Meanwhile, B cells produce various antibodies that target neoantigens on the tumor cell surface, aiming for tumor cell elimination, in contrast with Bregs that exert a suppressing effect on innate immune cells [47]. 

The neoantigens are also recognized by CD8+ T cells, via the interaction between the MHC-I tumor-associated antigens and T-cell receptor (TCR), and are associated with a more favorable prognosis, while dendritic cells (DCs), which are part of TME, facilitate the identification of the neoantigen CD8+ T cells. Tregs constitute a subpopulation of T cells that promote tumor progression, via interacting with TME components, by the expression of FOXP3 that suppresses the CD8+ T cells, as well as via releasing IL-2 that leads to NK cells’ deregulation [48]. Likewise, Bregs are also a subpopulation of cells that promote tumor cell progression via suppressing several immune cells such as T effector cells via releasing IL-10 or TGF-b, which constitute anti-inflammatory cytokines [49].

Meanwhile, the CD4-positive T cells produce a variety of molecules such as proinflammatory cytokines, while NK cells promote tumor progression limitation also via releasing cytokines in the systemic circulation (indirectly) or directly (cytotoxic granules), aiming for cancer cell destruction [50]. Additionally, CAFs, which constitute converted fibroblasts, are implicated in ECM alterations, while they either serve the role of tumor promoters or suppressors [51]. Neutrophils also present a dual role, primarily as suppressors of tumor growth, and, later, act as promoters (tumor-associated neutrophils (TANs)) of neoangiogenesis and local invasion via releasing VEGF and MMP-9, respectively [52], while another source of several types of metalloproteinases that also lead to ECM modification is adipocytes [53]. TAMs are another TME cell population that includes M1 and M2 subtypes that suppress and promote tumor progression, respectively [54]. Moreover, ECs promote neoangiogenesis via producing VEGF for the reassurance of malignant tumor blood supply, while they are also converted into CAFs [55]. Last but not least, MDSCs also have a tumorigenic and immunosuppressive role, as well as via increasing the expression of chemokine (C-C motif) ligand 2 (CCL2) and Chemokine (C-C motif) ligand 5 (CCL5) molecules [56]. 

#### 4.1.2. The Implication of TME in CCA Progression

The main histological characteristic of CCA is desmoplasia, which includes the presence of a distinctive fibrotic stroma, where CCA TME components and tumor cells lie. As was described above, TME components are implicated in CCA progression, by suppressing the anti-tumorigenic immune responses, via promoting tumor growth and local invasion, migration, as well as the neoangiogenesis, lymphangiogenesis, and metastatic dissemination of tumor cells [57]. The TME of CCA has a key role in the progression of this malignancy, as well as in its chemoresistance [58]. In the desmoplastic malignancies, including CCA, there are plenty of ECM modifications via the overproduction of several molecules that enhance the dense consistency of the stroma [59]. Some of the components that are neosynthesized are matrix metalloproteinases (MMPS) and periostin, as well as tenascin-C and osteopontin. However, osteopontin has a binary role in CCA progression, with its low levels promoting lymphatic dissemination, leading to an unfavorable prognosis [60]. Periostin overexpression is correlated with poor prognosis, as it promotes tumor growth, proliferation, and dissemination, while tenascin-C facilitates the interactions between the components of the matrix, promoting local invasion and metastasis via the EGFR and c-MET pathway [60,61,62]. Meanwhile, TAM overexpression is closely related to a worrisome prognosis and metastasis via enhancing the expression of hypoxia-inducible factor-1 (HIF-1α) that suppresses the T cells’ anti-cancer immune responses [63]. Additionally, when macrophages are activated by biliary tract injury, they release several proinflammatory cytokines that promote the proliferation of biliary tract cells, fibrogenic injury, and carcinogenesis via activating several molecular pathways such as JKN and Wnt/b-catenin, as well as neoangiogenesis via VEGF release [63,64]. TIL subpopulations have different locations and can influence the prognosis in different ways, such as the CD8-positive one at the core of the tumor and the CD4-positive one at the vicinity of the tumor bulk. Furthermore, CAFs lead to the reduction of anti-tumor immunity via interacting with DCs, preventing neoantigen presentation and T-cell activation, as well as pressing the proliferation of MDSC and T cells, as was shown in the CCA experimental models [65]. 

#### 4.1.3. The Role of EVs in CCA-TME 

As was described above, TME is a dynamic system, where several instances of intercellular cross-talk take place. More particularly, stromal elements closely interact with CCA cells in different ways, such as through the paracrine production of cytokines, enzymes, growth factors, or via cell-to-cell interactions. However, these are not the only ways that the CCA tumor interacts with its microenvironment, such as by releasing EVs that contain several cargoes, which have a variety of modulatory effects on the recipient cells [66]. There is increasing attention being paid to the implication of exosomes in the TME of several cancers, including CCA. We will further demonstrate the implication of EVs on the TME of this malignancy, based on the cargo they carry. 

(1)CCA-Derived EVs with Non-Coding RNA Molecules as Cargoes

Among the different cargoes that EVs can possibly carry, short or long non-coding RNA molecules are the most widely studied. There are several quantity and quality aberrations in EVs in the case of active malignancies, including CCA. Some types of EVs, carrying specific non-coding RNA molecules, are upregulated in CCA patients, compared with the healthy control groups, which can further lead to several modulations in TME via interacting with different recipient cells. There are several EV-carried microRNAs (miRNAs or miRs) that are highly up- or downregulated in CCA tumors, which can potentially be utilized as druggable targets and diagnostic or prognostic biomarkers [67]. Exosomes that carry miR-34c are highly regulated in CCA, leading to the promotion of tumor development, through interacting with CAFs [68]. On the other hand, there are some downregulated exosomal miRNAs identified that have an anti-tumorigenic effect on CCA, such as miR-30e and miR-195, constituting possible druggable targets [69,70]. The former suppresses EMT via targeting Snail, preventing tumor invasion and metastatic dissemination, while the latter also limits CCA growth and progression as was observed in the CCA culture model, which included the co-culture of the LX2 HSCs and HuCCT1 CCA cell lines [69,70]. Moreover, exosomal miR-200b-3p and miR-200c-3p are expressed in higher levels in CCA and they are proportionally related to the stage of tumor, implying their potential utilization as diagnostic/prognostic biomarkers [71]. Exosomal miR-183-5p is also tumorigenic, as it enhances programmed death-ligand 1 (PD-L1) overexpression in the macrophages, which is an immune checkpoint inhibitor that promotes iCCA development via the PTEN-AKT-PD-L1 signaling pathway [72]. Additionally, exosomal miR-183-5p also targets 5-hydroxy prostaglandin dehydrogenase (HPGD) in mast cells and CCA cells, leading to the upregulation of protumorigenic prostaglandin E2 (PGE2) and prostaglandin E receptor 1 (PTGER1) stimulation, in addition to inducing the release of VEGF from mast cells, promoting neoangiogenesis [73].

Furthermore, some other non-coding RNA molecules are circular RNAs (circRNA), which protect RNA molecules from the effect of degradative enzymes; however, it was recently demonstrated that they can also be protein-coding molecules [74]. Some exosomal circRNAs have been identified that can modulate CCA TME, leading to tumor progression, including Circ-0000284 [75] and Circ-CCAC1 [76]. More particularly, the former promotes the transformation of cholangiocytes into malignant ones, as well as CCA progression via the miR-637/LY6E pathway, inducing the overexpression of LY6E for the sponging of miR-637 [74,75]. 

The latter is highly expressed in malignant tissue or bile-derived EVs and interferes with the normal function of endothelial cells, promoting the neoangiogenesis, migration, and metastatic dissemination of tumor cells via upregulating Yin Yang 1 (YY1) through miR-514a-5p sponging [74,76]. 

(2)CCA-Derived EVs with Protein Molecules as Cargoes

There are many other cancer-related EVs that carry several molecules, including proteins such as vitronectin and integrin a/b, as well as lactadherin and frizzled class receptor 10 (FZD10). The former three proteins as EV cargoes are implicated in the upregulation of β-catenin, which leads to the increased migratory and invasive behavior of CCA, while the latter is involved in tumor reoccurrence, its metastatic dissemination, and increased CCA cell proliferation [66,77,78]. Another study by Z. Liu et al. (2022) demonstrated the implication of CCA-derived EVs carrying BMI1 (B-cell-specific Moloney murine leukemia virus integration site 1), which is a protein that induces cell proliferation and progression in many tumors, including CCA. Ev-BMI1 is highly found in CCA tissues, suppressing chemokine recruitment by T cells (CD8+), resulting in the post-translational modification (ubiquitination) of histones, such as H2A, which have a key role in the regulation of chromatin’s structure in the nucleus and the expression of genomic information. Meanwhile, these modifications lead to CCA growth, migration, local invasion, and metastatic dissemination, although BMI1 can be potentially utilized as a druggable target by its knockdown as a therapeutic option for several tumors, including CCA [79]. 

(3)Other CCA-Derived EVs

Moreover, it has been demonstrated by B. Oliviero et al. (2023) that EVs that are originating from iCCA have all the sphingolipid (SPL) lowered. However, it was noted that EVs secreted by iCCA with poor differentiation have increased dihydroceramide and ceramide levels, with the former being related to vascular system spreading and the latter being implicated in monocyte pro-inflammatory cytokine secretion, promoting iCCA progression and vascular invasion [80]. It is also observed that EVs that are derived from CCA cells can also modulate another TME element, the MSCs. More particularly, they increase the migratory behavior of MSCs, while they can also induce alterations in the tumor stroma by increasing the MSC’s conversion into CAFs, leading to desmoplasia. Meanwhile, they can also induce mRNA overexpression for several pro-inflammatory cytokines such as CXCL-1, a-SMA, and CCL2, as well as IL-6. Moreover, the exposure of MSCs to these CCA-derived EVs can lead to the increased cytokine release of interleukin-6(IL6), which acts as a growth factor for this malignancy, as well as to the upregulation of STAT-3 phosphorylation, leading to increased CCA cell multiplication [81,82]. As Haga et al. (2015) observed in their study, the exposure of MSCs KMBC or HuCCT1-derived EVs led to increased protein levels, including FAP and vimentin, as well as a-SMA. Moreover, Haga et al. (2015) also observed that the levels of some other proteins in MSCs, including those of chemokine (C-X-C motif) ligand 1 (CXCL-1), IL-6, and chemokine (C-C motif) ligand 2 (CCL2), were increased by up to 10.2 times when MSCs were exposure to KMBC-EVs and the uptake of their cargoes [82]. Last but not least, other studies observed that CCA-derived cells can modulate the expression levels of cytokine-induced killer cells (CIKs) in peripheral blood mononuclear cell cultures, leading to the decreased release of perforin and TNF-a, resulting in the phenomenon of tumor immune escape [83]. 

(4)TME Modulation by Different Sources of EVs

Some other sources of EVs that can modulate the CCA TME are HSCs and TAMs, more particularly, HSC-derived EVs that contain miR-195 as tumor suppressors, as they induce the inhibition of tumor growth and progression [84], while TAM-derived EVs containing circ-0020256 have a role as tumor promoters by increasing the proliferative, migratory, and metastatic behavior of the CCA cells [85]. In Table 1, we summarize the type of EV sources, cargoes, and their effect in TME, as well as their expression levels and role in CCA. Additionally, in Figure 3, we present a schematic presentation of the wide variety of TME components and tumor-promoting EV-cargoes that modulate CCA-TME (Agreement number RM25ZQJVSG).

### 4.2. The Implication of Parasite-Derived EVs in CCA Progression

As was previously referred to, liver flukes are well-studied carcinogens (group 1 carcinogens) that can potentially lead to cholangiocarcinogenesis and CCA development. As we already know, the persistence of this parasitic infection induces chronic inflammation in the biliary epithelium, which is attributed to the chronic exposure to parasite-secreted proteins [86,87]. However, the way that the parasite enters the cholangiocytes is not well-studied yet. As there is growing attention being paid on the emerging role of EVs’ role in cholangiocarcinogenesis, it is important to understand the ways that parasite-derived EVs can potentially lead to cholangiocyte transformation and tumorigenesis [88]. Moreover, EVs were firstly identified in infected bile samples from the aspiration of resected gallbladders and from infected animal models (hamsters), as well as in a culture medium with flukes, that, once they were internalized by cholangiocytes, led to an increased IL-6 cytokine release and upregulation of cell proliferation. However, the incubation and exposure of *O. viverrini* EVs with mouse anti–*O. viverrini* TSP-1 serum (EV recombinant tetraspanin (TSP)) suppressed the release of IL6, as well as the further uptake of EVs by cholangiocytes. Additionally, the uptake of parasite-derived EVs lead to several modifications in the wound-healing process, as well as carcinogenesis via the dysregulation of several types of protein expression and the presence of different isoforms, such as for PAK-2 ZO-2 and tropomyosin, respectively [89]. 

Furthermore, chronic infection by *Clonorchis sinensis (C. sinensis)* also appears quite injurious for the biliary epithelium, inducing chronic cholangitis, as well as fibrosis and CCA development. The EVs that are released by *C. sinensis* (CsEVs) have proven to be “key players” for the development of biliary injury via transferring Csi-let-7a-5p miRNA as cargoes, which are internalized by M1-type macrophages. The aforementioned miRNA induced the inhibition of Clec7a (dectin-1) and Socs1 (suppressor of cytokine signaling 1) that regulate the nuclear factor-kappa B (NF-κB) pathway, which is activated in pro-inflammatory states, leading to the release of several chemokines, cytokines, and surface receptors. More particularly, the former constitutes a receptor on the macrophage surface, which regulates macrophage polarization, while the latter regulates the balance between the M1 and M2 types of macrophages, aiming for a decrease of the pro-inflammatory signaling [90]. Moreover, *C. sinensis* infection induces the overexpression of TLR4 in the liver and enhances fibrosis in the biliary tract via the NF- κB-TGF-b-TLR4 signaling pathway. In the study of Y. Wang et al. (2023) [91], CsEVs were studied for the expression of TLR9, which constitutes a toll-like receptor-9 that recognizes unmethylated CpG motifs and activates the signaling pathway for immune response induction. More particularly, it was proven that CsEVs induce the upregulation of several regulatory proteins that have a key role in the main inflammatory signaling pathways such as ERK, AKT, p65, and p38 in the biliary epithelial cells (BECs), which were co-cultured with CsEVs. Additionally, a CsEV-induced oversecretion of TNF-a and IL6 cytokines via TLR9 expression in BECs was also observed [91]. Furthermore, Y. Wang et al. (2023) have demonstrated that the deletion of toll-like receptor-3 (TLR3) leads to severe clonorchiasis, the exacerbation of fibrosis, and high levels of proinflammatory cytokines such as TNF and IL-6 in BECs. However, fibrosis was diminished when BECs were expressing TLR3, via suppressing the expression of the aforementioned cytokines [92].

On top of all that, it was demonstrated that granulin, which is secreted by the O. *viverrini* fluke worm, acts like a growth factor for CCA cell lines via the induction of intercellular crosstalk. More particularly, cross-talk is achieved via EV-contained RNAs that are secreted by the CCA lines and they can modulate the MAPK phosphorylation in naïve cholangiocytes [93]. In Table 2, we demonstrate the role of EVs derived from fluke worms in the CCA progression.

## 5. EVs as Diagnostic Tools in CCA—A Novel Liquid Biopsy Approach

Despite the presence of the CA-19-9 serological biomarker, as a method for the early identification of CCA, it has limited sensitivity. There is a growing interest in EVs as diagnostic tools, which can potentially be combined with the already widely used CA-19-9 and can augment the accuracy of the diagnosis of this malignancy in its early stages. The presence of EVs can be found in different types of biological samples including blood, bile, saliva, ascitic fluid, and urine, as well as in seminal fluid and breast milk [94]. However, their presence is not only found in pathological conditions, as they have a key role in cell-to-cell communication, transferring several proteins, lipids, and metabolites for the optimal regulation of several homeostatic and immune-related pathways. With better knowledge of the non-diseased and the diseased EV cargo types, as well as of their quantity aberrations, which are observed between physiological and pathological conditions, we can potentially utilize EVs as diagnostic and screening tools [95].

### 5.1. EV as Diagnostic Tools in Bile Samples

It has been demonstrated that several EV-contained miRNAs in the bile of CCA patients can potentially be utilized as diagnostic biomarkers. More particularly, L. Li et al. (2014) identified 137 different miRNA species in 60 samples of bile. Among these miRNAs, cel-miR-39 had similarly elevated concentrations, while some other miRNAs, such as miR-126, miR-486-3p, miR-222, and miR-19a, had a null expression in the majority of the controls (biliary obstruction, PSC, and bile leak syndrome). However, the latter group of miRNAs were significantly increased in CCA patients’ bile, while miR-31, mir-484, miR-1274b, miR-16, miR-618, and miR-19a, as well as miR-486-3p and miR-191, were slightly increased in the controls, but more elevated in the CCA patients [96]. Moreover, another study by Severino et al. (2017) observed that the bile EV concentration was notably elevated (>10-fold) in pancreatic cancer and CCA patients, showing higher diagnostic accuracy than CA-19-9, while they are accurately classifying non-malignant (cholelithiasis) from malignant stenosis cases [84,97]. However, it was reported that one stenotic lesion that was characterized as benign based on the threshold of 9.46 × 10 nanoparticles/L in the bile sample was proven to be a malignant one, implying the necessity of further studies for the development of bile EV concentration levels as diagnostic and differentiating tools of malignant stenosis [97,98].

Furthermore, an upregulation of several long-non-coding RNAs has also been identified in CCA patients, such as ENST00000517758 and ENST00000588480.1, compared to the patients with obstructive pathology [99].

### 5.2. EVs in Blood Samples

Several aberrations have been demonstrated, more particularly in the quality than in the quantity of the serum EVs of CCA patients, in comparison with the controls [84,100]. The study by S.K Urban et al. (2020) demonstrated the synergistic effect of circulating serum EV and A-fetoprotein (AFP) levels in the differentiation of hepatobiliary malignancies with other types of cancers. More particularly, EVs-AnnV+CD44v6+ EVs showed auspicious results for biliary malignancy differentiation from HCC, with a 69% specificity and a high sensitivity of 91%, which constitute a combination of neoantigens. Additionally, combining the serum AFP values presented more favorable results (100% sensitivity and specificity) for biliary cancer differentiation from HCC [101]. Another study by X. Gu et al. (2020) reported that CCA patients have modified levels of piwi-interacting RNA (piRNA) in their plasma, in comparison with gallbladder carcinoma (GBC) patients and healthy patients. More particularly, CBC and CCA plasma samples had increased concentration levels of 323 piRNAs and 694 piRNAs, respectively. However, some piRNAs were also downregulated compared to healthy controls, such as 191 in GBC and 36 in CCA patients. Despite the fact that some piRNA types were found in both malignancies, they observed that there were some exosomal piRNAs that had different expression levels between CCA and GBC, which open up the horizons for CCA differentiation, implying that there are distinct signatures for each malignancy. More specifically, in the plasma of CCA and GBC patients, an upregulated plasma-circulating exosomal piR-10506469 level is identified, while in surgically-treated patients, the levels of exosomal piR-20548188 and piR-10506469 are notably limited, in comparison to pre-surgical concentration levels [102]. In CCA, they observed several downregulated plasma exosomal piRNAs such as nov-piR-2002537, nov-piR-14022777, nov-piR-17802142, nov-piR-4813367, and nov-piR-12355115, as well as nov-piR-15024555, nov-piR-9052713, nov-piR-4262304, nov-piR-3659538, and nov-piR-5114107. Meanwhile, they also observed the upregulation of exosomal piRNAs, especially before surgical CCA treatment, such as nov-piR-4333713 and nov-piR-10506469, as well as piR-20548188 and nov-piR-14090389 [102]. Additionally, the study by X-Y. Xue et al. (2020) demonstrated several down- or upregulated EV-contained miRNAs in blood samples of CCA patients. The former category of EV-miRNAs is found notably decreased in patients with CCA, including miR-218-5p, NC_000010.11_20947, miR-433-3p, and miR-9-3p, as well as miR-129-5p and NC_000001.11_1920, which were also decreased in GBC [103]. Meanwhile, in the case of the latter, they reported high expression levels of plasma exosomal-miRNAs, including exosomal miR-151a-5p and miR-4732-3p, as well as miR-191-5p and miR-96-5p, compared to healthy controls, especially in the II stages, implying their potential role as early diagnostic biomarkers. However, the aforementioned levels were significantly decreased in CCA patients 7 days post-surgery [103].

Furthermore, the study by A. Lapitz et al. (2023) reported several proteomic aberrations of serum EVs in patients with PSC-associated CCA patients versus healthy or PSC patients, implying their possible role as diagnostic, prognostic, and predictive serological biomarkers for CCA [104]. As was concluded by the machine-learning-based model, fibrinogen, ferritin light chain protein (FRIL), and C-reactive protein (CRP), all with/without the combination of carbohydrate antigen 19-9 (CA19-9), for the diagnosis of local PSC-associated CCA versus PSC alone, had an AUC of 0.947, which also enhanced the value of CA-19-9 as a biomarker alone. Meanwhile, they demonstrated the potential role of polymeric immunoglobulin receptor (PIGR), CRP, FRIL and fibrinogen as predictive biomarkers for CCA development in patients with PSC, before it is clinically evident. For the diagnosis of pan–CCA, FRIL and CRP had an AUC of 0.941, while for the diagnosis between non-PSC versus healthy controls, the Von Willebrand factor (VWF), CRP, and PIGR had an AUC 0.992. Finally, they reported, based on their multi-parametric analysis, that EVs can be also utilized as biomarkers for CCA prognosis, with PF4V, alpha-actinin-1(ACTN1), and MYC target 1 (MYCT1) proteins being related to a good prognosis and survival, and complement factor I (CFAI), cartilage oligomeric matrix protein (COMP), and G protein subunit i2 (GNAI2) with poor ones [104].

There are other studies about the proteomic analysis of serum EVs in patients with CCA, PSC, and HCC, compared to healthy controls. More particularly, several quality aberrations have been reported in CCA, associated with the type of EV cargoes in patients with CCA, implying their possible utilization as diagnostic, as well as staging, tools. Examples of EV cargoes that are highly expressed in CCA, in comparison with PSC alone, HCC patients, or healthy individuals, include fibrinogen gamma chain (FIBG), alpha-1-acid glycoprotein 1 (A1AG1), pantetheinase (VNN1), gamma-glutamyltranspeptidase 1 (GGT1), and CRP, as well as immunoglobulin heavy constant alpha 1 (IGHA1) proteins. Based on the aforementioned, all these EV cargoes can potentially help in the differential diagnosis between CCA cases, from healthy individuals to individuals with other malignancies or inflammatory diseases [67,105], whereas it has been also demonstrated that the presence of overexpressed EV cargoes, such as CRP, vitamin D-binding protein (VTDB), FIBG, and A1AG1, can be potentially utilized as differential diagnostic biomarkers between iCCA and HCC [67,105].

Last but not least, the transcriptomic analysis of serum EVs from CCA versus PSC patients demonstrated several mRNA transcripts as potential biomarkers for CCA differentiation from PSC, including phosphoglycerate dehydrogenase (PHGDH), activating transcription factor 4 (ATF4), and paraoxonase 1 (PON1), presenting an optimal AUC value of 1.00 as the differentiation method for CCA versus PSC patients [106].

### 5.3. EVs in Urine Samples and CCA Tissues

Another study by A. Lapitz et al. (2020) identified 26,066 RNAs in urine EVs of CCA patients, while the transcriptomic profiling of these urine EVs demonstrated alterations between healthy or PSC and CCA patients, as well as between CCA and ulcerative colitis or PSC. They concluded that the most auspicious biomarkers for CCA are the EV-containing Ras-related GTP binding D (RRAGD), MAP6-domain-containing 1 (MAP6D1), and INO80 complex subunit D (INO80D) [106]. Meanwhile, as it previously referred to the tumorigenic effect of EV-BMI1, the presence of high levels of EV-BMI1 in CCA tissues is considered an independent prognostic factor, closely related to an unfavorable prognosis, and increased CCA growth and invasion, as well as metastatic dissemination [79].

## 6. EVs s Staging, Prognostic, and Predictive Biomarkers

There have been several EV cargoes that can potentially be utilized as serum predictive biomarkers, including PIGR, CRP, FRIL, and fibrinogen, that predict CCA development in PSC patients before it is clinically evident [104]. Additionally, bile EV-MiR-183-5p increases PD-L1 expression, inducing anti-tumor immune suppression, and predicts the immune tolerance of iCCA [72].

Several studies have demonstrated the potential diagnostic role of several serum-circulating EVs for differentiating CCA from PSC. However, they can potentially give information about the stage of CCA in the early or advanced stages of the disease. EV-related proteins including FIBG, serum amyloid P-component (SAMP), ficolin-2 (FCN2), and plasma protease C1 inhibitor (IC1), as well as inter-alpha-trypsin inhibitor heavy chain H4 (ITIH4), are found highly expressed in CCA patients in stages I–II, in comparison to PSC patients, whereas high levels of EV-FCN2 and IC1 have an increased diagnostic value, compared to serum CA-19-9 in CCA patients in stages I–II [67,105]. The potential role of several exosomal miRNAs has also been identified not only as diagnostic but also as staging tools, including miR-96-5p, miR-4732-3p, and miR-151a-5p, as well as miR-191-5p, which are upregulated in CCA patients, particularly in stage II, compared to healthy controls, whereas patients with a higher stage of the disease, such as in stage III, mainly present high levels of exosomal miR-192-5p and miR-182-5p, as well as miR-191-5p [103].

However, the EV cargoes can also be used as prognostic tools, including good prognostic tools such as the ones with PF4V, ACTN1, and MYCT1 in serum [104], while there are several others that are associated with a poor prognosis, such as CFAI, COMP, GNAI2 [104], miR-200b-3p, and miR-200c-3p in serum, with the latter two having proportional levels to the tumor stage [71]. Last but not least, BMI1 [79] and miR-182/183-5p [72,73] have also been demonstrated as poor prognostic tools in CCA tissue. In Table 3, we demonstrate a summary of EV-based approaches for diagnosis, prognosis, and prediction in CCA.

## 7. EVs as Therapeutic Tools

EVs constitute carriers of a wide variety of molecules, which also offer protection from their enzymatic degradation. These vesicles are characterized as ideal drug vectors, which can deliver several chemotherapeutic compounds or miRNA molecules to the cancer site.

Exosomes as drug vectors have several advantages including not only their natural origin and biocompatibility, but also their decreased immunogenicity and their tendency to affect a particular organ system, based on the disease [107].

### 7.1. EVs as Drug Vectors

An example of EV-based therapy is the delivery of EV-miR-195 in CCA cells in vitro, which induces the suppression of cell growth via downregulating the expression of several pivotal proteins and growth factors such as cyclin-dependent kinases (CDK) 6, 1, and 4, which take part in cell cycle regulation, as well as VEGF. Moreover, based on the study of Li et al. (2017) The administration of LX2 (hepatic stellate cell line)-derived EVs that are loaded with an artificially embedded miR-195 ( highly downregulated miRNA in CCA) induce the suppression of tumor growth and the elongation of survival of the animal CCA model (rat), via limiting the desmoplastic reaction of stromal cells [70,108]. In addition, they observed that a-SMA and Ki67 expression levels were notably decreased in the rats in which an injection of EVs-miR-195 was administrated, which implies the anti-tumor effect of these EVs by suppressing CCA growth and progression, as well as the desmoplastic reaction [70].

Furthermore, another study by Ota et al. (2018) demonstrated the potential anti-tumor effect of EV-contained miR-30e for the suppression of CCA growth and progression, after they transfected HuCCT1 cells with miR-30e. More particularly, they took into consideration the fact that miR-30e that targets Snail (EMT-inducible transcription factor) is quite reduced in CCA, which was downregulated by TGF-B, leading to EMT modification. However, they observed that miR-30e overexpression led to EMT suppression and the inhibition of CCA cell migration by an miR-30e-enriched EV-induced Snail inhibition manner [69,109].

On top of that, another study by Chen et al. (2022) demonstrated the potential therapeutic effect of MSC-derived exosomes artificially loaded with the chemotherapeutic agent 5-fluorouracil (5-FU) in CCA cells in vitro. These EVs-5FU proved to be more effective in CCA cell elimination compared to free-5FU, implying their potentially effective use as drug vectors for this agent, which is a widely used chemotherapeutic modality [110].

### 7.2. EV-Based Vaccination

Moreover, the utilization of EV-based intraperitoneal vaccination has been demonstrated in animal models in the study by Chaiyadet S et al. (2019). More particularly, they performed vaccination of hamsters with *O. viverrini* EVs and with recombinant EV tetraspanin (surface protein) as an adjuvant, which encodes Ov-TSP-3 (rOv-TSP-3) and Ov-TSP-2 (rOv-TSP-2), as well as targets bile duct cells. It has been observed that vaccination-induced antibody production after cholangiocytes received the EV cargoes led to a response to the parasitic infection challenge. The mechanism of the response to the *O. viverrini* challenge is via the tetraspanin *O. viverrini* antibodies, which block the uptake and internalization of *O. viverrini*-EVs by the host cholangiocytes, resulting in a decrease of the *O. viverrini* burden, including the decline of worm growth, being notably shorter [111].

### 7.3. EV as a Therapeutic Target

Selective internal radiotherapy (SIRT) constitutes a quite effective therapeutic approach that is selected for patients with inoperable tumors in the late stage of the disease. In this therapeutic modality, several radioactive materials are used, which are delivered via an arterial catheter, directly into the tumor vasculature, leading to tumor necrosis. The study by F. Haag et al. (2022) demonstrated the utilization of SIRT for the modification of EV immune profiling in patients with inoperable CCA. They included 47 CCA patients who were receiving SIRT. They isolated EVs from patients prior to SIRT, as well as after the SIRT application, while they also performed EV phenotyping for the identification of the parental cells (such as immune-cell-derived EVs), including the identification of several markers such as CD4, CD8, CD44, CD40, CD49e, and MHCII. They observed aberrations between the EV origin before and after SIRT, such as the B-cell-derived EVs, which had notably elevated levels in the pre-treatment samples vs. the controls, as well as platelet-derived EVs (CD41b, CD62P, and CD42a as markers). However, in the case of the latter, they observed a significant decrease after the treatment, reaching the levels of the controls, but still being significantly higher than the controls post-SIRT. Moreover, they observed significantly increased or decreased markers in pre-treatment EVs, such as CD8, CD4, CD24, and CD44, as well as CD40 and MHCII, respectively. The levels of CD44 and CD24 were notably modified after the SIRT, as they were significantly reduced in the EV samples after the treatment application, implying the possible reduction of tumor growth and progression [112]. Moreover, they also demonstrated a decrease in CD133 marker expression after SIRT, which is found in EV-derived tumor progenitors and stem cells, implying the possible anti-tumor effect by modulating the release of EVs by the aforementioned parental cells. In conclusion, this study demonstrated the possible favorable effects of SIRT in modulating a wide variety of cell types and their EVs, including adaptive/innate immunity, as well as vasculature and PLT regulation [112].

Additionally, as was previously referred to, EV-BMI1 in CCA tissues/cells, promote CCA growth and progression, whereas BMI1 knockdown can lead to tumor growth suppression, via enhancing the anti-tumor immunity towards CCA cells and the immune checkpoint inhibition [79].

Furthermore, EV-contained ceramides and dihydroceramide that are derived from iCCA with poor differentiation, as mentioned earlier, are found in high concentration levels, inducing the increased secretion of pro-inflammatory cytokines by monocytes, leading to the vascular spreading of the disease and CCA growth. A serine palimotoyl transferase inhibitor has been utilized for the suppression of ceramide production, the so-called myriocin, which suppresses ceramide-induced inflammation [80]. Last but not least, it is demonstrated that phosphatase and tensin homolog (PTEN) deficiency induces exosome secretion, via its implication in the lysosome-related degradation process for MVBs, including the deregulation of transcription factor EB (TFEB), which constitutes a pivotal regulator of the lysosomal biogenetic pathway. It is reported that, when the PTEN’s crucial role for TFEB phosphorylation at Ser211 is lost, MVB degradation is inhibited and exosome secretion is increased, leading to CCA growth, invasion, and progression, as well as metastatic dissemination and recurrence. A potent therapeutic strategy for this deficiency is the administration of TFEB agonist curcumin analog C1, with the aim that the biogenetic pathway of exosomes could open new horizons for the prevention of CCA metastatic dissemination and cancer relapse [113]. In Table 4, we provide a summary of the EV-based therapeutic modalities.

## 8. Challenges and Limitations of EV Utilization for Research

EV-based therapeutic approaches open up new horizons for the treatment of several diseases, including CCA. The utilization of exosomes is in the spotlight of the current studies; however, there are only a limited number of clinical trials, due to several obstacles associated with EVs, which have to be overcome [114]. More particularly, there are several limitations in the handling of EVs such as their non-standardized isolation method, storage, and the cycles of freezing–thawing, which can induce alterations in EV behavior and functions [115]. The proper maintenance of EVs requires them to be stored at −80 °C, by which the water inside them becomes crystallized [116]. However, thawing is needed for the preparation of EV-related experiments or for the isolation of their contents. The cycles of freeze–thaw induce breakage of the vesicles and the release of their contents [117], so the protocol of EV handling needs to be appropriate for the maintenance of their integrity and functionality. This is the primary challenge for EV researchers, due to the fact that the freeze–thaw cycles may induce alterations in the EV functionality of inducing or modifying the translation of the target’s cell translation, which is achieved via the transfer of functional mRNA molecules that induce protein synthesis in the recipient cells [114,117,118]. A promising solution for the aforementioned issue is the utilization of freeze-dried EVs, which have decreased storage requirements and an increased viability period, as well as lower shipping expenses, due to their preservation in higher temperatures, such as room temperature [119]. During the process of freeze-drying, special antifreeze is added in order to protect the biomolecules, such as trehalose, which can successfully prevent the aggregation of the vesicles, which is commonly seen in freeze–thawing cycles. It is reported that exosomes that have undergone lyophilization did not have any significant difference in integrity and functionality with the ones that are preserved at −80 °C [120]. Finally, another technique for preservation is spray-drying, which is a procedure whereby the isolated EVs inside the liquid solution are transformed into a more stable dry form, the so-called powdered exosomes [120,121]. In conclusion, all the current limitations in the handling, storage, and transport of EVs can potentially be overcome via the utilization of several other storage techniques rather than cryopreservation, which can achieve enhanced usability for these emerging diagnostic and therapeutic tools.

## 9. Future Approaches in Diagnosis and Therapeutic Management of CCA

The study of the rising role of EVs is considered to be quite a promising area of scientific research, including their manipulation and application as diagnostic biomarkers and druggable targets, as well as drug vectors. There are several ongoing works of research on their utilization as diagnostic and prognostic tools for several diseases, including CCA, which open up the horizons for EV-based diagnostic modalities with higher specificity and sensitivity. Meanwhile, with further identification of EVs’ role in intercellular communication, their origin, and the sites of their uptake, we can potentially generate new prospects for their role as drug vectors for the suppression of tumor growth and progression, as well as for EMT modification. There is growing interest in the role of exosomes as a drug delivery system for several therapeutic agents, as well as for gene therapy, including the expansion of the field of EV engineering, by which EVs can be loaded by a wide variety of cargoes. Last but not least, they have a pivotal role in the immune system modulation for several autoimmune or malignant diseases, such as PSC and CCA, respectively, while their role in regenerative medicine is also in the spotlight of current studies, as they can potentially modify several signaling pathways, inducing tissue regeneration and repair. Currently, EVs are studied as potential biomarkers, capable of differentiating different pathologies, such as PSC, HCC, and CCA, from healthy individuals. It is pivotal to identify all the patients with PSC who will eventually develop CCA and identify it in the early stages for its optimal management, which can also be combined with minimally invasive diagnostic procedures such as spyglass choledochoscopy. However, there are several limitations such as the high-volume manufacturing of engineered exosomes, in which several drugs are embedded, implying the necessity of optimizing the production of EV drug-loaded exosomes in order to be potentially used in clinical practice. Lastly, another challenge for this potent drug delivery system is to explore its efficacy, as well as to underline its superiority versus the already existing nanoscale-based delivery systems, including polymeric and liposomal nanoparticles.

## 10. Conclusions

CCA is a highly malignant and heterogenic tumor, which is usually diagnosed in already advanced stages of the disease. The higher incidence of CCA over the last decades and the late disease detection, as well as CCA chemoresistance, necessitate the development of novel diagnostic tools that can allow the detection of CCA before it is clinically evident in high-risk groups of patients, such as those with PSC, as well as the development of novel drug-delivery systems and targeted therapies. EVs constitute nanoparticles of high significance, as they actively take part in intercellular communication by transferring a wide variety of substances that can regulate or deregulate the functions of the recipient cells. Having a better insight into the origin of EVs, and their role in intercellular communication and carcinogenesis will open up new horizons for the development of novel diagnostic and therapeutic modalities.

## Figures and Tables

**Figure 1 ijms-24-15563-f001:**
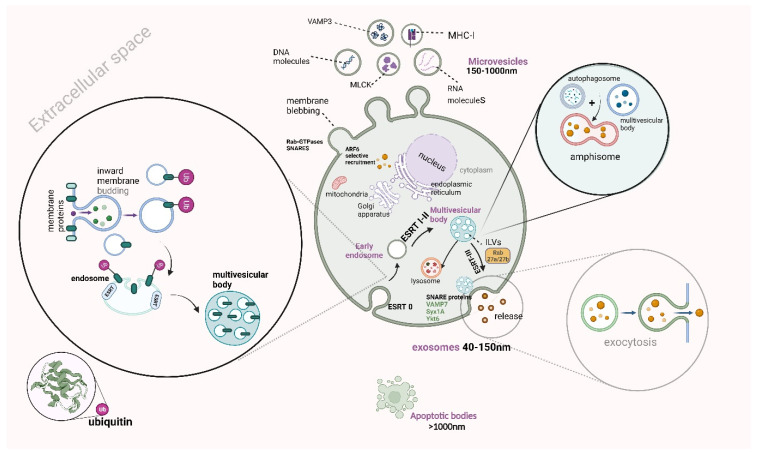
A schematic presentation of different routes of EV biogenesis. Exosomes are produced via the inward budding of the plasma membrane, resulting in the internalization of proteins under the effect of the endoplasmic network, which leads to the formation. The vesicles are split from the cell membrane, forming the early endosomes, which are further matured into late endosomes. Later, intraluminal vesicles (ILVs) are formed, which give rise to multivesicular bodies (MVBs) that fuse with the plasma membrane, releasing the exosome into extracellular space. However, MVBs can fuse with autophagosomes, resulting in amphisomes, which are either transferred to lysosomes for degradation or fused with plasma membrane for exosome exocytosis. Microvesicles are produced via the outward blebbing of the plasma membrane, while apoptotic bodies result from the cell apoptosis process, leading to apoptotic cell body generation, which gives rise to apoptotic bodies after their fragmentation. This figure was created with “BioRender.com”, accessed on 19 October 2023 (Agreement number UL25ZPCH0P).

**Figure 2 ijms-24-15563-f002:**
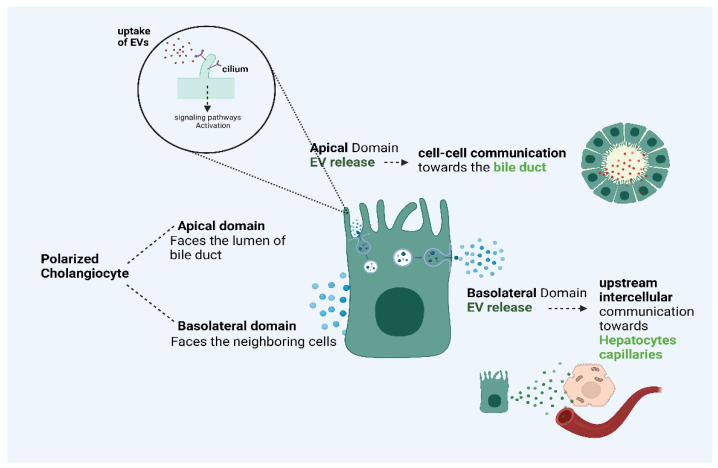
A schematic presentation of apical and basolateral secretion of EVs by polarized cholangiocytes and their targets. The polarized cholangiocytes release several EVs from their apical membrane region towards the bile duct and from their basolateral part towards hepatocytes and the capillaries around the intrahepatic ducts [43]. The apical EV release is implicated in cell–cell communication and induces several signaling pathways via EV uptake by cilia, while the basolateral EV release is implicated in upstream intercellular communication [43,44]. This figure was created with “BioRender.com”, accessed on 19 October 2023 (Agreement number XU25ZQ4AUV).

**Figure 3 ijms-24-15563-f003:**
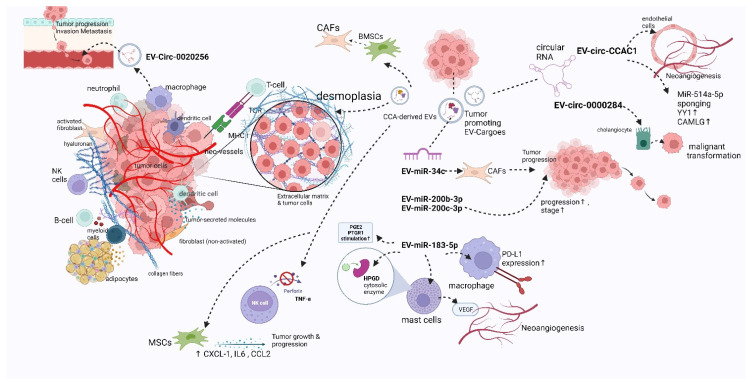
A schematic presentation of the wide variety of TME components and tumor-promoting EV cargoes that modulate CCA-TME. TME is a dynamic, heterogeneous system that comprises a wide variety of cells, including immune cells and adipocytes, as well as extracellular matrix, collagen fibers, neo-vessels, tumor-secreting-molecules, and chemokines. EVs are secreted by CCA cells, as well as other several cells of TME such as TAMs. This figure was created with “BioRender.com”, accessed on 19 October 2023 (Agreement number RM25ZQJVSG). (BMSCs) bone marrow mesenchymal stem cells, (Circ-CCAC1) cholangiocarcinoma-associated circular RNA 1, (YY1) Yin Yang 1, (CAMLG) calcium modulating ligand, (CXCL1) C-X-C motif ligand 1, (CCL2) chemokine (C-C motif) ligand 2, (IL-6) interleukin-6, (CAFs) cancer-associated fibroblasts, (HPGD) 5-hydroxy prostaglandin dehydrogenase, ↑ upregulated.

**Table 1 ijms-24-15563-t001:** The effect of EV components in TME and CCA progression.

Source/EVs Cargoes	Effect in TME	Role in CCA	Expression Levels
CCA cells			
MiR-34c [68]	CAF activation, tumor development	Tumor promoter	Upregulated
MiR-30e [69]	EMT suppression/targeting Snail/ Suppression of tumor invasion/metastatic dissemination	Tumor suppressor	Downregulated
miR-195 [70]	Limitation of tumor growth/progression (cell culture model)	Tumor suppressor	Downregulated
MiR-200b-3p MiR-200c-3p [71]	↑ tumor stage, growth, diagnostic biomarker	Tumor promoter Tumor promoter	Upregulated Upregulated
miR-183-5p [72,73]	iCCA progression via PD-L1 overexpression in macrophages aiming at HPGD in mast cells and CCA cells ↑ (PGE2) and PTGER1 stimulation ↑ VEGF release by mast cells, promoting neoangiogenesis	Tumor promoter	Upregulated
Circ-0000284 [75]	Cholangiocyte malignant transformation miR-637/LY6E pathway	Tumor promoter	Upregulated
Circ-CCAC1 [76]	Interacting with endothelial cells/ Promotion of neoangiogenesis Tumor growth/migration Sponging the miR-514a-5p/↑ YY1, ↑ CAMLG	Tumor promoter	Upregulated
FZD10 [66,77,78]	Proliferation/migration/metastatic dissemination Cancer reoccurrence	Tumor promoter	Upregulated
vitronectin, integrin a/b, lactadherin [66,77,78]; BMI1 [79]; ceramide [80], dihydroceramide	β-catenin overexpression Tumor invasion/migration Suppressed T-cell (CD8+) recruitment Ubiquitination of histones CCA growth, migration, local invasion, and metastatic dissemination Monocyte pro-inflammatory cytokine secretion Vascular spreading-invasion/Progression of iCCA	Tumor promoter Tumor promoter Tumor promoter	Upregulated Upregulated Upregulated
CCA-derived exosomes [81,82,83]	↑ MSC migratory behavior Formation of desmoplastic tumor stroma (↑ FAP, a-SMA, vimentin mRNA expression/Differentiation of BMSCs into CAFs to promote tumor stroma) ↑ CXCL-1, ↑ IL-6, ↑ CCL2 expression in MSCs in exposure of KMBC-EVs ↓ cytokine-induced killer cells (CIKs), ↓ perforin/TNF-a Tumor immune escape	Tumor promoter	Upregulated
HSCs			
MiR-195 [84]	Inhibition of tumor growth	Tumor suppressor	Downregulated
TAMs			
Circ-0020256 [85]	Proliferation/migration/Metastatic dissemination	Tumor promoter	Upregulated

(LY6E) lymphocyte antigen 6 family member E, (BMSCs) bone marrow mesenchymal stem cells, (FAP) fibroblast activation protein alpha, (a-SMA) alpha-smooth muscle actin, (Circ-CCAC1) cholangiocarcinoma-associated circular RNA 1, (YY1) Yin Yang 1, (CAMLG) calcium-modulating ligand, (FZD10) frizzled class receptor 10, (CXCL1) C-X-C motif ligand 1, (CCL2) chemokine (C-C motif) ligand 2, (IL-6) interleukin-6, ↑ upregulated, ↓ downregulated.

**Table 2 ijms-24-15563-t002:** The role of EVs derived from fluke worms in CCA progression.

Fluke-Worm-Derived EVs	Role in CCA Progression
*O. viverrine*-derived EVs [89,93]	Modified wound-healing process Dysregulation of several protein expression ↑ IL-6 cytokine release/↑ cell proliferation Cholangiocarcinogenesis ↑ parasite-secreted granulin Intercellular crosstalk between CCA-secreted EVs and naïve cholangiocytes Modulation of MAPK phosphorylation in naïve cholangiocytes
*C. sinensi*s-derived EVs [90,91,92]	EV-contained Csi-let-7a-5p are internalized by M1-type macrophages. Inhibition of Clec7a and Socs1 Deregulation of NF-κΒ pathway ↑ proinflammatory state Overexpression of TLR4 ↑ biliary tract fibrosis via NF-κB-TGF-b-TLR4 signaling pathway Upregulation of ERK, AKT, p65, and p38 in BECs TNF-a, IL6 oversecretion via TLR9 expression in BECs

(*O. viverrini*) *Opistorchis viverrini*, (*C. sinensis*) *Clonorchis sinensis*, (BECs) biliary epithelial cells, (TLR9) toll-like receptor 9, (TLR3) toll–like receptor-3, ↑ upregulated.

**Table 3 ijms-24-15563-t003:** A summary of EV-based diagnostic, prognostic, and predictive biomarkers in CCA.

Type of Biomarker	Sample	Type of Cargoes	Expression in CCA Samples
Diagnostic	Bile	cel-miR-39 miR-126 miR-486-3p miR-222 miR-19a	↑ concentration in CCA patient’s bile vs. (null expression) PSC, bile leak syndrome, bile obstruction [96]
	Bile Bile Bile Serum	miR-31 mir-484, miR-1274b miR-16 miR-618 miR-19a, miR-486-3p miR-191 Bile-EVs long-non-coding RNAs ENST00000517758 ENST00000588480.1 circulating serum EVs and A-fetoprotein levels AnnV+CD44v6+ PiRNAs piR-10506469 PiRNAs piRNAs nov-piR-2002537 nov-piR-14022777 nov-piR-17802142 nov-piR-4813367 nov-piR-12355115 nov-piR-15024555 nov-piR-9052713 nov-piR-4262304 nov-piR-3659538 nov-piR-5114107. nov-piR-4333713 nov-piR-10506469, piR-20548188 nov-piR-14090389 nov-piR-10506469, piR-20548188 miR-218-5p NC_000010.11_20947 miR-433-3p miR-9-3p miR-129-5p NC_000001.11_1920 miR-151a-5p, miR-4732-3p miR-191-5p miR-96-5p Fibrinogen FRIL CRP FRIL/CRP VWF/CRP/PIGR FIBG A1AG1 VNN1 GGT1 CRP IGHA1 CRP, VTDB FIBG A1AG1 mRNA transcripts PHGDH ATF4 PON1 miR-200b-3p miR-200c-3p RRAGD MAP6D1 INO80D	↑ concentration in CCA patient’s bile vs (lower expression) PSC, bile leak syndrome, bile obstruction [96] ↑ over 10-fold increased levels in CCA and pancreatic cancer Higher diagnostic accuracy than CA19-9 Differentiation of malignant stenosis vs. cholelithiasis Threshold of 9.46 × 10 nanoparticles/L in the bile sample [84,97,98] ↑ in malignant stenosis vs. obstructive pathology [99] Synergistic effect of serum EVs and AFP differentiation of hepatobiliary malignancies vs. other types of cancers [101] Differentiation of biliary malignancies vs. HCC ↑ diagnostic value by combining AFP [101] ↑ concentration levels of 323 EV-piRNAs in GBC /694 piRNAs CCA vs. healthy [102] ↓ concentration levels of 191 EV-piRNAs in GBC/36 in CCA vs. healthy [102] Downregulated only in CCA [102] Upregulated in CCA (↑ pre-surgical levels) [102] ↓ post-surgical exosomal piRNA concentration levels in CCA [102] Downregulated EV-miRNAs in CCA/GBC vs. healthy [102] Upregulated EV-miRNAs in CCA/GBC vs. healthy (↑ stage II CCA) [103] ↓ post-surgical levels (7 days after) +/− CA-19-9 for diagnosis of local PSC-associated CCA vs. PSC ↑ in Pan-CCA vs. healthy Non-PSC vs. healthy [104] ↑ in CCA vs. (PSC, HCC, healthy) [67,105] ↑ in iCCA vs. HCC [67,105] ↑ in CCA vs. PSC patients [106] ↑ in CCA—early detection of CCA proportional to tumor-stage [71] Most auspicious urine EV biomarkers for CCA [106]
	Serum Plasma Plasma Plasma Plasma Serum Serum Serum Serum Serum Serum		
Predictive	Serum Bile	PIGR/CRP/ FRIL/Fibrinogen MiR-183-5p	Predictive biomarkers for CCA development in patients with PSC, before it is clinically evident [104] ↑ PD-L1-expression-immune suppression Immune tolerance of ICC [72]
Staging	Serum	FIBG SAMP FCN2 IC1 ITIH4 miR-96-5p miR-4732-3p miR-151a-5p miR-191-5p	Overexpressed in I–II stage vs. PSC [67,68,69,70,71,72,73,74,75,76,77,78,79,80,81,82,83,84,85,86,87,88,89,90,91,92,93,94,95,96,97,98,99,100,101,102,103,104,105] Overexpressed in II stages vs. healthy [103]
	Serum	miR-192-5p miR-182-5p miR-191-5p	Overexpressed in III stage [103]
Prognosis	Serum CCA Tissue Serum Tissue/bile	PF4V/ACTN1/MYCT1 CFAI/COMP/GNAI2 BM1 MiR-200 family miR-200b-3p miR-200c-3p miR-182/183-5p	Good prognosis-survival of CCA patients [104] Poor prognosis-survival of CCA patients [104] Independent poor prognostic biomarker [79] ↑ levels in CCA—↑ proportionally to tumor-stage [71] ↑ levels—poor prognosis [72,73]

(FRIL) ferritin light chain protein and (CRP) C-reactive protein, (CA19-9) carbohydrate antigen 19-9, (PIGR) polymeric immunoglobulin receptor, (VWF) Von Willebrand factor, (ACTN1) alpha-actinin-1, (MYCT1) MYC target 1 proteins, (CFAI) complement factor I, (COMP), G protein subunit i2 (GNAI2) cartilage oligomeric matrix protein, (FIBG) fibrinogen gamma chain, (A1AG1) alpha-1-acid glycoprotein 1, (VNN1) pantetheinase, (GGT1) gamma-glutamyltranspeptidase 1, (VTDB) vitamin D-binding protein, (PHGDH) phosphoglycerate dehydrogenase, (ATF4) activating transcription factor 4 and (PON1) paraoxonase 1, (RRAGD) Ras-related GTP binding D, (MAP6D1) MAP6-domain-containing 1, (INO80D) INO80 complex subunit D, (SAMP) serum amyloid P-component, (FCN2) ficolin-2, (IC1) plasma protease C1 inhibitor, (ITIH4) inter-alpha-trypsin inhibitor heavy chain H4, (BMI1) B-cell-specific Moloney murine leukemia virus integration site 1, ↑ upregulated, ↓ downregulated.

**Table 4 ijms-24-15563-t004:** EV-based therapeutic modalities.

Modality	Cargoes	Anti-Tumor Effect
EV as a drug delivery system	MiR-195 [70,108]	Suppression of CCA cells in vitro Downregulation of CDK6, CDK1, CDK4, VEGF Transfected LX2 (hepatic stellate cell line) Suppression of tumor growth Elongation of survival of the animal CCA model (rat) via limiting the desmoplastic reaction of stromal cells ↓ a-SMA and Ki67 expression levels in the rats
	MiR-30e [69,109]	Transfected HuCCT1 cells with miR-30e Suppression of CCA growth and progression miR-30e that targets Snail (EMT-inducible transcription factor) EMT suppression and inhibition via Snail inhibition manner
	F-FU [110]	MSC-derived exosomes artificially loaded with 5-FU in vitro EVs-5FU proved to be more effective in CCA cell elimination Vs free-5FU
Vaccination SIRT-targeting EVs [112] EV as therapeutic target BMI1 knockdown Μyriocin TFEB agonist curcumin analog C1 [113]	Tetraspanins *O. viverrini* Evs [111]	The mechanism of the response to *O. viverrini* challenge is via the tetraspanin *O. viverrini* antibodies block the uptake and internalization of *O. viverrini* EVs by host cholangiocytes Decrease of the *O. viverrini* burden, ↓ worm growth (shorter)
	BMI1 [79] Ceramide [80]	Selected for patients with inoperable tumors Radioactive materials are delivered via an arterial catheter, directly into the tumor vasculature SIRT for the modification of EV immune profiling ↓ PLT-derived EVs (CD41b, CD62P and CD42a) post-treatment ↓ CD44 and CD24—EVs after the SIRT ↓ tumor growth and progression. ↓ EV derived by tumor progenitors and stem cells CD133 after SIRT CCA tumor progression/growth Enhancing the ICI effect Suppression of ceramide production Ceramide-induced inflammation Suppression of vascular invasion Aiming at the biogenetic pathway of exosomes Prevention of CCA metastatic dissemination and cancer relapse

↓ downregulated.

## Data Availability

Not applicable.

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
