# Peer review of "The Arising Role of Extracellular Vesicles in Cholangiocarcinoma: A Rundown of the Current Knowledge Regarding Diagnostic and Therapeutic Approaches"

_ijms, 2023, doi:10.3390/ijms242115563_

Round 1

Reviewer 1 Report

Trifylli et al. have undertaken a commendable effort in summarizing the multifaceted role of extracellular vesicles (EVs) in Cholangiocarcinoma (CCA). While the paper commendably sheds light on the potential of EVs in influencing disease progression and their uses as diagnostics and therapeutics, a few enhancements could accentuate the manuscript's focus on EVs:

1. In the Introduction part, briefly introduce the current diagnostic and therapeutic approaches for CCA, prior to diving into the vast world of EVs. Highlighting the challenges or drawbacks associated with these traditional methods can serve as a prelude to elucidate the potential advantages of EV-based approaches.

2. Given the intricate nature of the section 3, adding a visual representation could greatly benefit the reader. Consider incorporating a figure that schematically delineates the role of EVs in biliary tract physiology and cholangiopathies, which is the foundational aspects of this subject.

3. To keep in line with the focus of the manuscript, Figure 2 needs to be revised to accentuate the role of EVs. While the heterogeneity of the Tumor Microenvironment (TME) is undeniably important, the primary focus should remain on EVs, given the scope of this review.

4. On line 642, under the title of section 7.2, the term "Ev-based vaccination" seems to have a minor typographical error. Kindly change "Ev" to "EV" for consistency. On line 722, in the concluding paragraph of section 8, the phrase "such as the High-volume" needs a minor adjustment. The capital "H" in "High-volume" is unnecessary and should be rendered in lowercase for correctness.

Author Response

Firstly, the authors would like to thank Reviewer #1 for his/her constructive suggestions for our manuscript. Please see below for detailed responses to his/her minor/major comments.

  1. In the Introduction part, briefly introduce the current diagnostic and therapeutic approaches for CCA, prior to diving into the vast world of EVs. Highlighting the challenges or drawbacks associated with these traditional methods can serve as a prelude to elucidate the potential advantages of EV-based approaches.

Reply: We added current diagnostic and therapeutic approaches, as well as the limitations for these modalities (lines 75-93).

  1. Given the intricate nature of the section 3, adding a visual representation could greatly benefit the reader. Consider incorporating a figure that schematically delineates the role of EVs in biliary tract physiology and cholangiopathies, which is the foundational aspects of this subject.

Reply: We added Figure 2. in the revised manuscript as you proposed.

  1. To keep in line with the focus of the manuscript, Figure 2 needs to be revised to accentuate the role of EVs. While the heterogeneity of the Tumor Microenvironment (TME) is undeniably important, the primary focus should remain on EVs, given the scope of this review.

Reply: We revised Figure 3. (previously Figure 2) as you proposed.

  1. On line 642, under the title of section 7.2, the term "Ev-based vaccination" seems to have a minor typographical error. Kindly change "Ev" to "EV" for consistency. On line 722, in the concluding paragraph of section 8, the phrase "such as the High-volume" needs a minor adjustment. The capital "H" in "High-volume" is unnecessary and should be rendered in lowercase for correctness.

Reply: We corrected the word EV and the high-volume as you suggested.

Reviewer 2 Report

COMMENTS:

1.     Author should review the whole article again for minor grammatical errors (For example, “auspisious” should be corrected to “auspicious”, DNA hype-methylation to DNA hyper-methylation) or awkward phrasing that may disrupt the flow of reading. (line: 57, 83-84, 94-96, 97, 131, 206,…..)

2.     The author should be consistent, for example, use “Biliary” or “biliary “consistently throughout the text.

3.     In the 4th section “The implication of EVs in CCA”, author should provide a brief overview of TME and its significance in CCA progression. This will help readers to understand it easily.

4.     The text would benefit from simplifying some sentences for clarity. For example, the sentence, "It is important to demonstrate all the current information about the ways that parasite-derived EVs can potentially lead to the malignant transformation of cholangiocytes and tumorigenesis" could be revised to something like, "We need to understand how parasite-derived EVs contribute to cholangiocyte transformation and tumorigenesis."

5.     Author should include the publication year, for example- you have mentioned that the studies by B. Oliviero, Haga et al., and Y. Wang et al., but it’s unclear that when these studies were conducted.

6.     The section about using EVs for diagnosis could be made more reader-friendly by clearly indicating how EVs are used for diagnosis and what specific markers are associated with CCA.

7.     Author should follow the rules of binomial nomenclature while writing the scientific names (line: 69-70).

8.     There are sentences that are quite long and complex, which can make it challenging for readers to follow. Please try to break them down into shorter, more digestible sentences.

9.     The figure 1, EVs biogenesis pathway is very simple, need to make it more attractive, involving of ESRT and other important proteins for microvesicle formation.

10.   “Section “4.2. The implication of Parasite-derived EVs in CCA progression” needs to be removed. It is meshing up.

11.  I would suggest to include separate section of challenges and Limitation on EVs to be utilized for the research. Please cite Karn et al., 2021 https://www.mdpi.com/2227-9059/9/10/1373.

12.  One additional figure is required to show how EVs microRNA involved in CCA via different pathways.

Minor English corrections are required. 

Author Response

  1. Author should review the whole article again for minor grammatical errors (For example, “auspisious” should be corrected to “auspicious”, DNA hype-methylation to DNA hyper-methylation) or awkward phrasing that may disrupt the flow of reading. (line: 57, 83-84, 94-96, 97, 131, 206,..)

Reply: Firstly, the authors would like to thank Reviewer #2 for his/her constructive suggestions for our manuscript. Please see below for detailed responses to his/her minor/major comments.

We performed English editing of the manuscript and we eliminated grammar and spelling errors.

We corrected the words auspicious, DNA hyper-methylation (line 57), and the phrases that Reviewer#2 suggested.  Additionally, we rephrased the sentences (in blue)

  • lines 106-109, “The interplay between EVs and cholangiocarcinogenesis has gained a considerable amount of attention, especially for the implication of EVs in the CCA growth and progression, as well as for their potential use as diagnostic and therapeutic tools [15].”
  • Lines 117-120

The heterogeneity of these nanoparticles is also identified in their cargoes, which are enclosed in their double-lipid membrane. These cargoes can be lipid, DNA, protein molecules, receptors, autophagosomes as well as coding and non-coding RNA molecules, including messenger and short    or long-non-coding RNA molecules, respectively.

  • Line 159: However, there are several other proteins that take part in membrane bledding such as Rab-GTPases and N-ethylmaleimide-sensitive factor attachment proteins (SNAP) receptors (SNARES), which induce cargo recruitment and microvesicle production under hypoxia [28].
  • Line 238:

 Meanwhile, there are several hepatocytes-derived EVs that are implicated in cholangiocarcinogenesis such as integrin beta-4 (ITGB4) and epidermal growth factor receptor (EGFR) [43,44]. 

  1. The author should be consistent, for example, use “Biliary” or “biliary “consistently throughout the text.

Reply: We replaced all the “Biliary” with “biliary”, which is consistently used in the manuscript.

  1. In the 4th section “The implication of EVs in CCA”, author should provide a brief overview of TME and its significance in CCA progression. This will help readers to understand it easily.

Reply: In section 4.1 The interplay between EVs and TME in CCA, we start with the subsection

4.1.1. A brief review of TME components and their role, where we provide all the basic information for TME components. Afterwards, in 4.1.2 we particularly describe the implication of TME components and their significance in CCA progression.

  1. The text would benefit from simplifying some sentences for clarity. For example, the sentence, "It is important to demonstrate all the current information about the ways that parasite-derived EVs can potentially lead to the malignant transformation of cholangiocytes and tumorigenesis" could be revised to something like, "We need to understand how parasite-derived EVs contribute to cholangiocyte transformation and tumorigenesis."

Reply: We simplified the sentence in Line 483 it is important to understand the ways that parasite-derived EVs can potentially lead to cholangiocyte transformation and tumorigenesis [88].

  1. Author should include the publication year, for example- you have mentioned the studies by B. Oliviero, Haga et al., and Y. Wang et al., but it’s unclear when these studies were conducted.

Reply: We added the year of publication in the parts where we mention several studies.

  1. The section about using EVs for diagnosis could be made more reader-friendly by clearly indicating how EVs are used for diagnosis and what specific markers are associated with CCA.

Reply: We modified the section as you suggested.

  1. The author should follow the rules of binomial nomenclature while writing the scientific names (line: 69-70).

Reply: We corrected the way of writing the scientific names based on the binomial nomenclature rules (larvae of Clonorchis sinensis and Opisthorchis viverrini in line 69-70, as well as in each part we talk about these parasites.

  1. There are sentences that are quite long and complex, which can make it challenging for readers to follow. Please try to break them down into shorter, more digestible sentences.

Reply: We followed your suggestions and broke the long sentences into shorter ones.

  1. The figure 1, EVs biogenesis pathway is very simple, need to make it more attractive, involving of ESRT and other important proteins for microvesicle formation.

Reply:  We added several pieces of information in figure 1.  about ESRT and other proteins involved in microvesicle formation as you recommended. (In the Revised Manuscript Figure 2.)

  1. “Section “4.2. The implication of Parasite-derived EVs in CCA progression” needs to be removed. It is meshing up.

 Reply: Parasitic infection with larvae of Clonorchis sinensis and Opisthorchis viverrini constitutes a pivotal risk factor for CCA development and a huge burden for Southeast Asia and China. Infection with liver flukes, via fresh water or by eating undercooked fish is common in Southeast Asian countries, resulting in the chronic inflammation of bile ducts and eventually in scaring and cholangiocarcinogenesis.  We truly believe that this section should remain, otherwise we exclude one of the most common risk factors of CCA development in Southeast Asian countries and a big portion of patients. The implication of the parasite-derived EVs in CCA progression is quite interesting and there are several new studies in the field of EVs and CCA ,however  the information is not currently summarized in any recent review.

  1. I would suggest to include separate section of challenges and Limitation on EVs to be utilized for the research. Please cite Karn et al., 2021 https://www.mdpi.com/2227-9059/9/10/1373.

Reply: We added a separate section of “Challenges and Limitations on EVs utilization for the research”.  (section 8 in the revised manuscript) as you suggested and cited the manuscript that you recommended to us (citation [114]).

  1. One additional figure is required to show how EVs microRNA involved in CCA via different pathways.

Reply: We revised Figure 2 (in the revised manuscript Figure 3, where we demonstrate several EV-miRNAs that are implicated in CCA progression. 

Round 2

Reviewer 1 Report

The manuscript is in a good format and ready for publication. 

Reviewer 2 Report

Authors addressed all my comments and manucripts has been improved now. Therefore, I recommend it for publication in IJMS journal.